# Comparative Study of Semen Parameters and Hormone Profile in Small-Spotted Catshark (*Scyliorhinus canicula*): Aquarium-Housed vs. Wild-Captured

**DOI:** 10.3390/ani11102884

**Published:** 2021-10-03

**Authors:** Marta Muñoz-Baquero, Francisco Marco-Jiménez, Ximo García-Domínguez, José Luis Ros-Santaella, Eliana Pintus, María Jiménez-Movilla, Daniel García-Párraga, Francisco Alberto García-Vazquez

**Affiliations:** 1Departamento de Producción y Sanidad Animal, Salud Pública Veterinaria y Ciencia y Tecnología de los Alimentos, Instituto de Ciencias Biomédicas, Facultad de Veterinaria, Universidad Cardenal Herrera-CEU, Calle Tirant lo Blanc, 7, 46115 Alfara del Patriarca, Spain; marta.munozbaquero@uchceu.es; 2Departamento de Fisiología, Facultad de Veterinaria, Campus de Excelencia Internacional Mare Nostrum, Universidad de Murcia, 30100 Murcia, Spain; 3Institut of Animal Science and Technology, Universitat Politècnica de València, 46022 Valencia, Spain; fmarco@dca.upv.es (F.M.-J.); ximo.garciadominguez@gmail.com (X.G.-D.); 4Department of Veterinary Sciences, Faculty of Agrobiology, Food and Natural Resources, Czech University of Life Sciences Prague, Kamýcká 129, 165 00 Praha-Suchdol, Czech Republic; rossantaella@gmail.com (J.L.R.-S.); eliana.pintus27@gmail.com (E.P.); 5Department of Cell Biology and Histology, Faculty of Medicine, University of Murcia, Campus Mare Nostrum and IMIB-Arrixaca, 30100 Murcia, Spain; mariajm@um.es; 6Veterinary Services, Avanqua-Oceanogràfic S.L 46013, Ciudad de las Artes y las Ciencias, C/Eduardo Primo Yúfera, 1B, 46013 Valencia, Spain; dgarcia@oceanografic.org; 7Research Department Fundación Oceanogràfic de la Comunitat Valenciana, Gran Vía Marqués del Turia 19, 46005 Valencia, Spain

**Keywords:** Scyliorhinidae, Elasmobranchii, spotted dogfish, sperm, conservation biology

## Abstract

**Simple Summary:**

Comprehensive knowledge of chondrichthyan reproductive biology is crucial for the development of reproductive technologies. For that reason, a male reproductive evaluation was performed on the basis of a comparison of samples collected from wild-captured and aquarium-housed small-spotted catshark (*Scyliorhinus canicula*). Semen quality, sperm morphometry, and reproductive hormones were assessed. The results demonstrate good in vitro semen quality in aquarium-housed sharks, although there was lower plasma testosterone.

**Abstract:**

Several chondrichthyan species are threatened, and we must increase our knowledge of their reproductive biology in order to establish assisted reproductive protocols for ex situ or in situ endangered species. The small-spotted catshark (*Scyliorhinus canicula*) is one of the most abundant shark species of the Mediterranean coast and is easy to maintain in aquaria; therefore, it is considered an ideal reproductive model. This study aimed to compare *S. canicula* male reproductive function in aquarium-housed (*n* = 7) and wild-captured animals, recently dead (*n* = 17). Aquarium-housed animals had lower semen volume (*p* = 0.005) and total sperm number (*p* = 0.006) than wild-captured animals, but similar sperm concentrations. In terms of sperm parameters, aquarium-housed sharks showed higher total sperm motility (*p* = 0.004), but no differences were observed regarding sperm viability, mitochondrial membrane potential, or membrane integrity. A morphometric study pointed to a significantly longer head (*p* = 0.005) and acrosome (*p* = 0.001) in wild-captured animals. The results of the spermatozoa morphological study of *S. canicula* were consistent with previous results obtained in other chondrichthyan species. With regard to sex hormones, testosterone levels were significantly lower in aquarium-housed animals (*p* ≤ 0.001), while similar levels of 17β-estradiol and progesterone were found. In short, the present study provides evidence of good in vitro semen quality in *S. canicula* housed in an aquarium, underlining their excellent potential for application in reproductive technologies for this and other chondrichthyan species.

## 1. Introduction

Human progress, constant population growth, and resource consumption are the leading causes for biodiversity loss and the increased number of threatened and endangered species [1,2]. Among Chondrichthyes species, sharks, rays, and chimaeras are facing a global conservation crisis, suffering population decline and possible extinction due to factors such as overfishing, pollution, habitat destruction, and climate change [3,4,5,6]. Of the 1226 species of sharks and rays [7] and 52 chimaera species [8] in the wild, it is estimated that more than one-quarter are threatened according to IUCN Red List criteria, due to overfishing (targeted and incidental), habitat degradation, and other pollution threads such as heavy metals, microplastics, or endocrine disruptors [9,10,11,12,13]. Chondrichthyans are described as “apex predators”, with few natural enemies and feeding on animals below them in the food chain, thereby stabilizing populations of other species [14] and playing a crucial role in the ecosystem [15]. Even though small-spotted catsharks (*Scyliorhinus canicula*) are an extremely abundant shark species globally [16], warning signs due to their sensitivity to overexploitation and pollution have already been reported [17,18,19].

Increasingly, zoos and aquaria are becoming essential for the conservation of endangered species, both directly through captive breeding programs and indirectly by improving our understanding of the biology, behavior, and reproduction of a given species [20]. It has been estimated that over 102 chondrichthyan species are kept in European zoos and public aquaria, accounting for 8.6% of all known species [10]. Chondrichthyan are charismatic and common animals at zoos and aquaria worldwide for exhibition, preservation, and learning purposes [21]. Aquarium breeding programs are useful tools to ensure healthy populations and improve husbandry standards [22]. However, most chondrichthyan reproduction in aquaria is based on natural mating [23], although their reproduction might benefit from the use of reproductive technologies (monitoring reproduction, sperm collection, artificial insemination, or sperm preservation). Several reproductive features have been studied in some chondrichthyan species, such as hormonal cycles [24,25,26], semen [27,28,29,30], and spermatozoa morphology [30,31,32] and morphometry [33,34]. However, new studies are required to compare the reproductive features between wild and aquarium-housed animals, so as to improve reproductive technologies for conservation purposes [23].

Given the limited data on the reproductive features of small-spotted catsharks (*Scyliorhinus canicula*), this study aimed to compare (1) semen parameters, and (2) testosterone, estradiol, and progesterone concentrations between aquarium-housed and wild-captured small-spotted catsharks.

## 2. Materials and Methods

### 2.1. Animals and Housing

A total of 18 semen samples were collected during the study period from seven aquarium-housed small-spotted catsharks at the Oceanogràfic of Valencia (Spain). They were all housed in a 5000 L closed system with UV light and ozone for disinfection, under controlled water quality conditions. A photoperiod of 12:12 h was kept throughout the experiment. Aquarium-housed males were kept isolated from females. The aquarium-housed animals were maintained under human care at the aquarium at least 1 year before the experiment started, being completely adapted to the aquarium environment. The water parameters in aquarium-housed animals were monitored throughout the experiments (17.0–21.0 °C, 5.1 mg/L oxygen, 36 g/L salinity, and 7.6–8.2 pH). The wild sharks (*n* = 17 obtaining 17 semen samples) were donated by local fisheries from the Region of Valencian Community (Spain), which were captured accidentally, and formed part of commercial and artisanal fisheries in Valencia (39°26′45″ N, 0°19′12″ W), Jávea (38°47′21″ N, 0°09′47″ E) and Cullera (39°09′58″ N, 0°15′10″ W) (Figure 1). The water parameters of the wild sharks were 14.6–19.0 °C and 34–37 g/L salinity, information obtained from Mediterranean Sea records from the Valencia buoy (39°52′ N, 0°20′ E) (Figure 1). From both groups, only animals classified as mature males, displaying rigid and fully calcified claspers [35], were used in the study. Moreover, a biometric analysis was performed, and total length (from the snout to the longest length of the tail, cm), width (circumference of the shark body at its maximum width, cm), clasper length (from the cloaca to the longest length of the claspers, cm), and weight (g) of each individual prior to semen collection were measured.

### 2.2. Experimental Design

A total of 24 animals were used in the present study, 17 wild-captured and seven aquarium-housed sharks. According to the goals of the study, several reproductive variables were compared from both experimental groups to establish the differences in weight (g) and length (cm), seminal volume (μL), spermatozoa concentration (×10^6^ spermatozoa/mL), total sperm number per sample (×10^6^ spermatozoa), total sperm motility (%), viability (%), mitochondrial membrane potential (%), membrane integrity (%), morphology, and morphometry. Individual blood samples from both experimental groups were collected for hormonal analysis (17β-estradiol (E2), progesterone (P4), and testosterone (T)). 

### 2.3. Sample Collection and Processing

Semen samples were collected from November 2019 to March 2020. Aquarium-housed animals were placed in dorsal recumbency, so-called “tonic immobility”, creating a mild sedation [36,37]. The posterior portion of the body was supported out of the water, and the cloacal area was wiped with a paper towel and rinsed with sterile shark Ringer’s solution based on the ionic composition of shark blood (22 g/L urea and 9 g/L NaCl) [38] in order to clean the surface and reduce bacterial contamination. Semen was collected by abdominal massage (stripping). Briefly, the semen samples were obtained by applying firm but gentle downward pressure on the ampulla with one finger, moving slowly toward the cloaca. Semen was collected directly from the urogenital papilla using a 5 mL syringe with care taken to avoid contamination with urine or feces. Visually contaminated samples were discarded. Blood samples were collected through caudal venipuncture (ventral coccygeal vein) using heparinized syringes coupled to 21-gauge needles and transferred to lithium–heparin tubes. Blood was centrifuged at 3000× *g* for 10 min to obtain the plasma, which was stored at −80 °C. After sample collection, sharks were released back into the quarantine tank, and their recovery was monitored by aquarium staff. Samples were transported to the laboratory within 1 h of collection due to the geographical proximity, and the assessment was performed immediately.

In the case of wild-captured sharks, samples were taken immediately when the fishery boat arrived at the port, 4–8 h post capture in the sea, depending on the daily fishery routines. The semen was recovered from recently dead animals as previously described, applying finger pressure and collecting all the sperm present in the ampulla. Blood was collected from the caudal vein following the same protocol as described above. Collected semen and blood samples were maintained in the dark at 4 °C and then transferred within 1 h to the laboratory facilities, and the evaluation was performed immediately upon arrival. The room temperature during the assessment procedure for both groups was 21–23 °C.

### 2.4. Seminal Quality Assessment

The semen quality variables studied were semen sample volume, sperm concentration, total sperm number, total motility, viability (DNA-binding fluorescent dye), mitochondrial membrane potential (JC-1 kit dye), and sperm membrane integrity (hypoosmotic swelling test). The volume of the semen sample was measured by an automatic micropipette. Aliquots from each sample were diluted 1:100 in shark Ringer’s solution to assess sperm concentration, and then 10 μL of the resulting solution was placed in a 10 μm deep Makler counting chamber. Aliquots from each semen sample (5 μL) were diluted 1:20 with shark Ringer’s solution to immediately assess total motility. Semen samples were analyzed using a phase-contrast microscope at 200× magnification (Nikon E 400). Total motility was estimated by the percentage of motile sperm, including spermatozoa vibrating without moving forward [21]. The same sample was assessed for the percentage of live and dead spermatozoa using the LIVE/DEAD sperm viability kit (ThermoFisher), which consists of two DNA-binding fluorescent stains: a membrane-permeant stain, SYBR-14, and a conventional dead-cell stain, propidium iodide (PI). The SYBR-14 stained the nuclei of living sperm in bright green, with an absorption spectrum of 488 nm and emission spectrum at 518 nm when bound to DNA, while PI stained only sperm that had lost their membrane functionality (Figure 2A). To obtain this dye, 1 μL of SYBR and 1 μL of PI were mixed with 100 μL of the diluted semen sample. The evaluation was performed under a fluorescent microscope (400× magnification) (Nikon E 400), evaluating at least 100 cells per sample. Another aliquot was assessed for mitochondrial membrane potential using the JC-1 assay kit (ThermoFisher). The fluorescent carbocyanine dye, JC-1, colors mitochondria with high membrane potential in orange and mitochondria with low membrane potential in green (Figure 2B). A stock solution of JC-1 (2 μL) was mixed with 100 μL of diluted sperm. An incubation period of 30 min in darkness was necessary to allow the reagent to stain the spermatozoa. Finally, the semen was assessed by placing the sample (5 μL) directly on a microscope slide. The evaluation was performed under a fluorescent microscope at 400× magnification (Nikon E 400), evaluating at least 100 cells per sample. Lastly, sperm membrane integrity was evaluated using the hypoosmotic swelling test (HOST). HOST evaluates the ability of cells to swell, indicating whether the membrane remains intact or not (Figure 2C). A hypoosmotic solution was prepared with shark Ringer’s solution at 500 mOsm/kg to induce sperm swelling after an incubation period of 10 min. The evaluation was performed under an optic microscope, 400× magnification (Nikon E 400), evaluating at least 100 cells per sample. 

### 2.5. Sperm Morphometrics and Morphology 

Sperm morphometry was analyzed according to previous studies with minor modifications [39]. Semen samples for each experimental group were pooled (four different animals) for morphometric evaluation. Collected sperm samples were diluted 1:50 with 2.5% glutaraldehyde (Sigma-Aldrich, Madrid, Spain) in a solution prepared with shark Ringer’s solution. A fixed sperm subsample was used to make the smears, which were air-dried for one day before mounting. Images (resolution 2560 × 1920 pixels, TIFF format) were taken using a digital camera (Digital Sight DSFi1, Nikon, Tokyo, Japan) under a phase-contrast microscope (Nikon Eclipse E600, Japan; 40× objective). Pixel size was 0.14 µm in the horizontal and vertical axes. Spermatozoa were assessed using *ImageJ* software (National Institutes of Health, Bethesda, MD, USA). The sperm morphometric parameters studied (35 spermatozoa in each group) were the acrosome, head, acrosome-head, and midpiece lengths (in μm). 

### 2.6. Transmission Electron Microscopy

Spermatozoa were fixed in 1.25% glutaraldehyde in phosphate-buffered saline (PBS, Sigma-Aldrich^®^, Madrid, Spain) (*v/v*, pH 7.4) and incubated at 4 °C for 2 h. After fixation, spermatozoa were post-fixed in potassium ferrocyanide reduced osmium tetroxide for 1 h. After extensive washing, the samples were then dehydrated through a graded series of ethanol and processed for embedding in Epon 812. Ultrathin sections were obtained with an ultramicrotome (Microm International GmbH) and mounted on coated nickel grids. Ultrathin sections were counterstained with uranyl acetate followed by lead citrate and imaged in a Philips Tecnai 12 transmission electron microscope.

### 2.7. Field-Emission Scanning Electron Microscopy (FE-SEM)

Spermatozoa were fixed in 1.25% glutaraldehyde in PBS (*v/v*) and washed three times in PBS. Following post-fixation using osmium tetroxide 1% with potassium ferricyanide for 2 h, specimens were washed in sodium cacodylate 0.1 M buffer with sucrose. Spermatozoa were dehydrated with acetone series and were critical point dried over Isopore filters 0.4 µm (Merck Millipore Ltd., Tokyo, Japan). Finally, specimens were platinum sputtered with a 5.0 nm thin layer (Leica EM ACE 600) and were examined using an FE-SEM, (ApreoS Lovac IML Thermofisher, Waltham, MA, USA), with a selected voltage of 5 kV. 

### 2.8. Sexual Hormonal Analysis

Blood samples were centrifuged at 3000× *g* for 10 min to obtain plasma and were placed in cryovials and stored in an ultralow freezer (−80 °C) for further sexual hormone analysis. Plasma was carefully pipetted in an ACCES 2 Immunoassay (Beckman Coulter, Brea, CA, USA) at the laboratory in the Oceanogràfic of València (Valencia, Spain). The levels of testosterone, progesterone, and 17β-estradiol in blood plasma were measured using the radioimmunoassay (RIA) method. Briefly, 100 μL of standards, controls, and plasma samples were transferred to microtubes. Calibration and quality control of the reagents were performed according to the manufacturer’s recommendation. Assays were validated prior to the study. The upper limits of detection for testosterone, progesterone, and 17β-estradiol were 16 ng/mL, 30 ng/mL, and 5200 pg/mL, respectively. Individual plasma samples were run at 1:10 dilution in 0.9% saline solution and were compared with testosterone, progesterone, and 17β-estradiol standards.

### 2.9. Statistical Analysis 

All statistical analyses were performed with the SPSS 21.0 software package (SPSS Inc., Chicago, IL, USA). The data obtained for the experimental variables were compared to ascertain statistically significant differences between aquarium-housed and wild-captured groups. Descriptive statistics were used for all the parameters. The assumption of normality and homogeneity of variances were evaluated by Shapiro–Wilk and Levene tests, respectively. When both tests were fulfilled, Student’s *t*-test was applied (biometric, spermatozoa concentration, morphometry data, and hormone values (testosterone, progesterone and 17β-estradiol)) between both groups. For those variables whose data were not normally distributed, the nonparametric Mann–Whitney U test was used (volume, total spermatozoa concentration, total motility, viability, mitochondrial membrane potential, and membrane integrity test). The level of significance was set at *p* < 0.05. Only total width (cm) was higher in aquarium-housed than in wild-captured animals (Table 1). Biometric data, volume (mL), sperm concentration, total concentration, and morphometrical data (acrosome length (μm), head length (μm), and midpiece length (μm)) were expressed as the mean ± standard deviation (SD) and coefficient of variation (CV) using the formula CV = (standard deviation/mean) × 100, while total motility (%), viability (%), mitochondrial membrane potential (%), and membrane integrity (%) were expressed as the mean ± standard error of the mean (SEM).

## 3. Results

### 3.1. Comparison of Biometric Parameters between Aquarium-Housed and Wild-Captured Individuals

Biometric data (weight (g), total length (cm), total width (cm), and clasper length (cm)) for the aquarium-housed animals and wild-captured animals did not show any significant differences (*p* > 0.05), and only total width differed, being greater in aquarium-housed animals (Table 1). Animals from the two experimental groups were considered homogeneous.

### 3.2. Comparison of Semen Parameters between Aquarium-Housed and Wild-Captured Individuals

Results for the semen parameters (volume, concentration, and total sperm number) are shown in Table 2. Sperm volume in aquarium-housed sharks was significantly lower than in wild-captured sharks (833.3 ± 422.87 vs. 1905.9 ± 1110.44 μL, *p* = 0.005). No differences were found in sperm concentration between aquarium-housed and wild-captured sharks (78.9 ± 43.36 vs. 107.8 ± 50.42 × 10^6^ spermatozoa/mL, *p* = 0.072), but there were differences in the total number of spermatozoa per sample, which was statistically lower in aquarium-housed than in wild-captured sharks (72.8 ± 68.27 vs. 219.6 ± 188.57 × 10^6^ spermatozoa, *p* = 0.006). The results for sperm functionality (total motility, viability, membrane integrity, and mitochondrial membrane potential) are shown in Table 2. Total sperm motility in aquarium-housed sharks was significantly higher than in wild-captured animals (48.7% ± 6.96% vs. 18.8% ± 4.15%, *p* = 0.004). However, no statistical differences were observed for the remaining parameters (sperm viability: aquarium-housed = 72.9% ± 5.55% vs. wild-captured = 78.3% ± 3.30%, *p* = 0.717), mitochondrial membrane potential (90.1% ± 6.97% vs. 96.1% ± 0.70%, *p* = 0.372) and membrane integrity (70.0% ± 6.75% vs. 72.9% ± 5.14%, *p* = 0.882).

### 3.3. Comparison of Sperm Morphology between Aquarium-Housed and Wild-Captured Individuals

The morphological study of the spermatozoa (acrosome, head, midpiece, and flagellum) was conducted using transmission electron microscopy and field-emission scanning electron microscopy (FE-SEM) (Figure 3). The acrosome in *S. canicula* had a helical shape and a spiral expansion that bent slightly, with a depression fitting over the tip of the nucleus and several ridges on the plasma membrane, termed the acrosome “crown”. A subacrosomal rod filled with heterogeneous material was observed longitudinally in the region between the acrosome and the nuclear tip. This posterior end of the acrosome was slanted in the cross-section, and several ridges were observed on the plasma membrane. The tip of the nucleus was covered by the parachromatin sheath and showed an attenuated cone shape. The posterior nuclear area was connected with the midpiece by the basal nuclear fossa and fitted the tip of the midpiece axial rod. The midpiece was surrounded by a fibrous sheath, protecting the interior polyhedral mitochondria, distributed around the complete length of this structure. Mitochondria were present as multiple concentric cristae. The main midpiece part was covered with the cytoplasmic sleeve, a double-membrane granular layer, coated with invaginated vesicles. The central midpiece structure was the axial rod, fitted into the posterior end of the nucleus. The complete midpiece was covered with a fibrous sheath that overlapped the posterior end of the midpiece and covered the anterior part of the spermatozoon tail, inserted into the distal centriole of the flagellum. A cytoplasm canal was observed between the fibrous sheath and the continuous base of the cytoplasmic sleeve. The distal centriole, a ring of fibrous material attached to the anterior end of the midpiece, extended posteriorly, covering the axoneme. The initial part of the axoneme was accompanied by two longitudinal accessory axonemal columns, with an oval cross-section and flattened interior surface, sometimes appearing in the shape of a kidney. Where the axonemal columns ended, the axoneme lay along the central axis of the spermatozoon. 

### 3.4. Morphometric Comparison of Sperm between Aquarium-Housed and Wild-Captured Individuals

All the evaluated parameters except midpiece length showed significant differences between aquarium-housed and wild-captured animals (Table 3). More specifically, acrosome (2.78 ± 0.466 vs. 3.01 ± 0.354 μm, *p* = 0.027), head (40.96 ± 1.113 vs. 41.79 ± 1.284 μm, *p* = 0.005), and acrosome-head (43.75 ± 1.153 vs. 44.80 ± 1.274 μm, *p* = 0.001) lengths were longer in wild-captured sharks, while midpiece length showed no statistical differences (18.16 ± 0.988 vs. 18.31 ± 1.139 μm, *p* = 0.566 for aquarium-housed sharks vs. wild-captured, respectively). 

### 3.5. Comparative Analysis of Hormonal Reproductive (Testosterone, Progesterone, and Estradiol) Profile between Aquarium-Housed and Wild-Captured Individuals

As can be seen in Figure 4, progesterone and estradiol concentrations were similar in aquarium-housed and wild-captured individuals (0.26 ± 0.21 ng/mL vs. 0.28 ± 0.17 ng/mL, *p* = 0.684; and 0.05 ± 0.04 ng/mL vs. 0.03 ± 0.02, *p* = 0.151, respectively), but the testosterone concentration was significantly higher in wild-captured animals (55.2 ± 24.25 ng/mL vs. 17.5 ± 10.30 ng/mL, *p* < 0.001).

## 4. Discussion

Several studies have revealed the worldwide decline in the populations of elasmobranch species over the past half-century, resulting in a clear risk of their extinction [40,41]. To prevent this crisis, assisted reproductive technologies need to be developed in aquarium-housed populations (reviewed by [42,43]). Understanding the basic reproductive biology of a species is critical for developing such technologies [21]. To the best of our knowledge, this is the first study to describe the semen parameters and the hormone profile of small-spotted catsharks (*Scyliorhinus canicula*), and it offers a comparison between wild-captured and aquarium-housed males. The findings of the study demonstrate that the semen quality of aquarium-housed and wild-captured sharks was broadly similar, although aquarium sharks had lower testosterone levels. 

Reproduction in aquaria is infrequent or even nonexistent in some species of sharks [10]. Although previous research has focused on chondrichthyan semen collection and sperm quality assessment for the monitoring of male reproductive function in aquarium-housed animals [44], to our best knowledge, only one study compared reproductive parameters in aquarium-housed and wild-captured *Carcharias taurus* sharks using only live animals [21]. It should be emphasized that, according to our biometric analysis, none of the variations observed in this study can be attributed to dissimilarities in maturity [45]. Aquarium-housed small-spotted catsharks showed a lower semen volume and total number of sperm, but higher total motility compared with wild-captured animals, which we attribute to the variations observed in the management of wild-captured sharks. In wild-captured animals, the semen was collected from recently dead animals, as the bycatch from trawling (4–8 h after death depending on the fishery daily routines), where muscles were relaxed, and the ampulla could be emptied. Abdominal massage or stripping, as occurs in aquarium-housed sharks, only releases mature spermatozoa together with seminal plasma [46]; however, when this technique is used in freshly dead sharks, it is possible to recover the vast majority of sperm present in the caudal reproductive tract. This fact could also explain the differences in motility between groups as wild-type semen samples comprised all the sperm forms contained on the ducts. Another alternative and potentially complementary explanation for the higher semen volume observed may be the sexual competition between wild-captured males and the post-copulatory sperm competition due to female mating attempts with multiple males [21,47]. Although the observation of lower total sperm motility in the wild-captured small-spotted catsharks was unexpected, previous studies have found that aquarium-housed and aquaculture fish have lower sperm motility than wild-captured fish [21,48,49,50]. In chondrichthyans, spermatozoa acquire motility after transiting through the epididymis and are stored in an initial motile state in the seminal vesicles, increasing the motility during their movement through the epididymal channel, with spermatozoa being fully motile after contact with seawater [28]. Furthermore, chondrichthyan spermatozoa, have several environmental influences, such as medium osmolality, which increases their velocity under the influence of uterine fluid in the female reproductive tract [27]. Furthermore, chondrichthyan sperm were highly active in a mixture of seminal plasma and fluid from the alkaline gland, suggesting that secretions from male secondary sexual organs may also be important for sperm activity [51]. Despite those semen peculiarities related to wild-captured sharks, sperm viability, mitochondrial membrane potential, and sperm membrane integrity were similar in aquarium-housed and wild-captured small-spotted catsharks, even after considering the possible effect of sample collection in recently deceased animals. Previous studies using SYBR-14/ PI staining described a higher percentage of spermatozoa to be alive than motile ones [21,44]. It is important to remark the importance of using recently dead animals accidentally captured as a gamete source, which allows the use of these valuable biological samples for further application in reproductive studies focused on the conservation of endangered species.

Small-spotted catshark spermatozoa were similar in morphology to other chondrichthyan species, being composed of a helically shaped head, midpiece with cytoplasmic sleeve, flagellum, and acrosome [21,32,44,52]. To the best of our knowledge, this is the first time that transmission electron microscopy images of spermatozoa have been studied in this species. The acrosome crown and subacrosomal rod filled with heterogeneous material was also identified as a perforatorium and was observed longitudinally in the region between the acrosome and the nuclear tip [32,33,53,54]. The anterior half of the acrosome of *S. canicula* spermatozoa bent from the sperm axis, and this characteristic was also found in *Hydrolagus colliei* [55]. In other chondrycthye species, as reviewed by Jamieson, the main midpiece part is covered by the cytoplasmic sleeve, and by a double-membrane granular layer coated with invaginated vesicles, where mitochondria were observed [32]; however, but in contrast to some species described (such as *Squalus suckleyi*), the midpiece in *S. canicula* was not helical. A cytoplasmic sleeve, the remnant of germ cell cytoplasm, was present in both aquarium-housed and wild-captured spermatozoa, sliding off the midpiece and running the whole flagellum length, similar to the mammal cytoplasmic droplet [21,56]. With regard to sperm morphometry, the length of the head and midpiece fell within the range described in other species, confirming that there is no relationship with the size of the species [44]. Nevertheless, sperm collected from aquarium small-spotted catsharks was found to have slightly smaller acrosome, head, and acrosome-head lengths than the sperm from wild-captured sharks. Several authors have associated the considerable intraspecific variation in spermatozoa morphology and morphometry with post-copulatory sexual selection and multiple paternity [30,32,57], which has been described in many chondrichthyans, including *S. canicula* [47]. Additional hypotheses include differences in terms of nutrition and other environmental factors. In this species, the temperature, not the photoperiod, is probably responsible for regulating the annual reproductive cycle [58]. Admittedly, the temperature in the aquaria in the study was similar to that found in the Mediterranean Sea during the experiment period (17.0–21.0 °C vs. 14.6–19.0 °C, aquarium and Mediterranean Sea (Valencia buoy: www.puertos.es/es-es/oceanografia/Paginas/portus.aspx accessed on 12 April 2020), respectively). Interestingly, testosterone and most of the other steroids peaked in winter and early spring in wild-captured *S. canicula* from the Mediterranean Sea, coinciding with the experimental period of the present study [25,59]. Previous studies showed that testosterone levels were lower in the aquarium-housed sharks [21,25]. However, our results point to similar levels of progesterone and 17β-estradiol. Even though previous research in aquarium sharks showed that low testosterone levels were related to poor reproductive response [44], our results suggest that the aquarium-housed and wild-captured small-spotted catshark possess a comparable semen quality in vitro. The effect of cortisol produced during the capture and death process could decrease androgen levels in wild fish [60], although other studies used samples from commercial sharks species after lethal techniques to analyze sexual hormones [61]. The present study found higher testosterone levels in wild-captured sharks similarly to other results reported in previous research using living animals [21]. However, further research is necessary to ascertain the in vivo sperm response within the female reproductive tract (e.g., after interaction with the uterine fluid) when artificial insemination is performed.

## 5. Conclusions

We are in the early steps of applying reproductive technologies in small-spotted catsharks, and a thorough knowledge will only be achieved after more research, which will allow their application in the future. The present study provides strong evidence of the good in vitro semen quality in small-spotted catsharks housed in an aquarium and confirms the excellent potential for applying reproductive technologies to manage this species. Nevertheless, further research is necessary to increase our knowledge for implementing successful breeding programs in aquaria. In particular, artificial insemination studies are required to confirm their potential use in sharks. 

## Figures and Tables

**Figure 1 animals-11-02884-f001:**
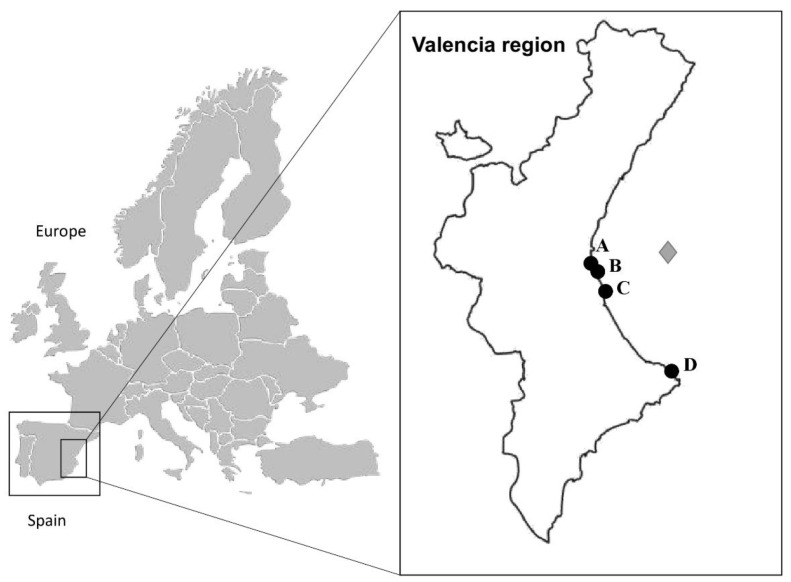
Geographical locations of aquarium and fisheries for the small-spotted catshark (*Scyliorhinus canicula*) used in this study. Maps of Europe, Spain, and Valencia (area inside the square). Black dots indicate the sample collection points. (**A**) Oceanogràfic de Valencia (aquarium-housed animals); (**B**–**D**) Valencia (39°26′45″ N, 0°19′12″ W), Cullera (39°09′58″ N, 0°15′10″ W), and Jávea Ports (38°47′21″ N 0°09′47″ E), respectively (wild-captured animals during fishing). The gray diamond indicates the Valencia buoy, (39°52′ N, 0°20′ E) used to monitor water parameters related to wild-captured animals (www.puertos.es/es-es/oceanografia/Paginas/portus.aspx, accessed 12 April 2020).

**Figure 2 animals-11-02884-f002:**
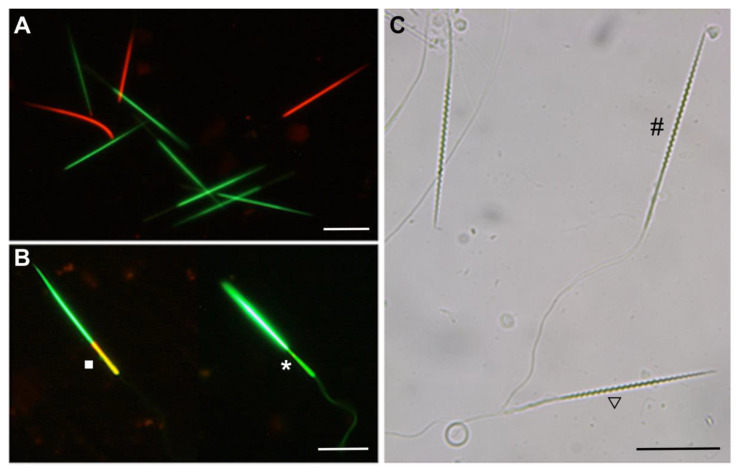
Representative images of small-spotted catshark (*Scyliorhinus canicula*) sperm cell quality. Scale bar, 20 μm. (**A**) Epifluorescence micrographs of sperm cells stained with SYBR-14 (green) and propidium iodide (red) at 200× magnification. Green fluorescence shows live sperm, and red fluorescent indicates dead sperm. (**B**) Epifluorescence micrographs of sperm cells stained with JC-1 at 200× magnification. The dye changes emission wavelength depending on membrane potential by a shift from orange, showing high potential (square), to green color, showing low potential (asterisk). (**C**) Phase-contrast image displaying tail swelling to assess the functional integrity of membrane by hypoosmotic swelling test at 200× magnification. Coiled tail spermatozoa were identified as having a functional intact plasma membrane (triangle); normal tail spermatozoa were identified as having a nonfunctional intact plasma membrane (octothorpe).

**Figure 3 animals-11-02884-f003:**
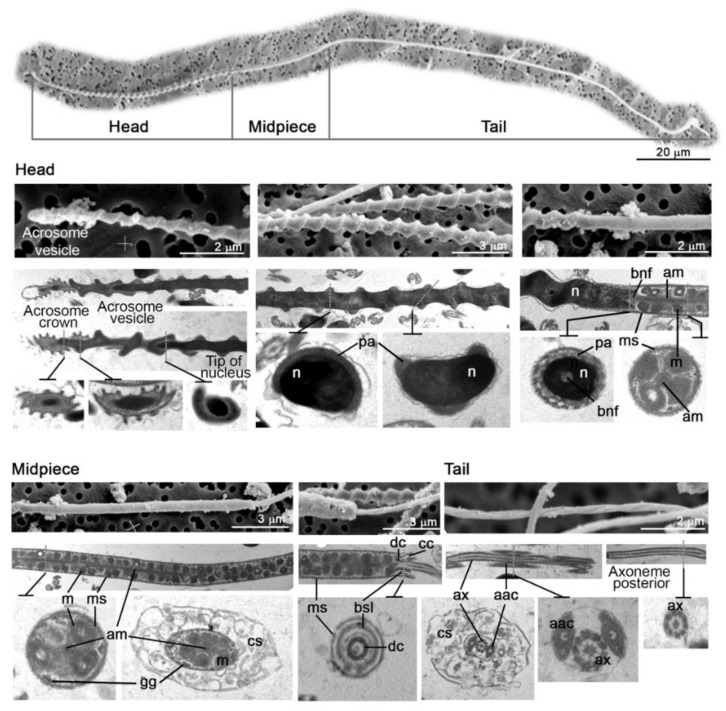
A representative image of ultrastructure of small-spotted catshark (*Scyliorhinus canicula*) sperm cells. Scanning and transmission electron micrographs of longitudinal and transverse sections through the main pieces of the spermatozoon. The top pictures were obtained with transmission electron microscopy, while the bottom pictures were obtained with scanning electron microscopy. The spermatozoa designed parts are labeled as follows: aac, accessory axonemal column; am, axial rod of midpiece; ax, axoneme; bnf, basal nuclear fossa; bsl, persistent base of cytoplasmic sleeve; cc, cytoplasmic canal; cs, cytoplasmic sleeve; dc, distal centriole; gg, glycogen granules; m, mitochondrion; ms, fibrous sheath of midpiece; n, nucleus; pa, parachromatin.

**Figure 4 animals-11-02884-f004:**
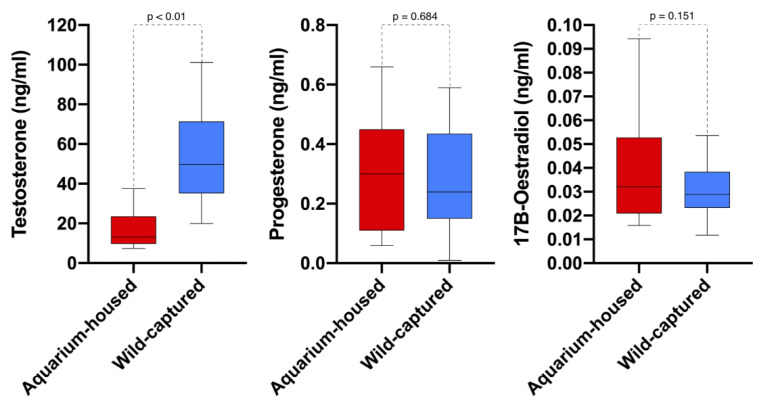
Box plot comparing blood hormonal sexual values between aquarium-housed and wild-captured small-spotted catshark (*Scyliorhinus canicula*).

**Table 1 animals-11-02884-t001:** Biometric comparison for aquarium-housed and wild captured small-spotted catshark (*Scyliorhinus canicula*) males. Data are expressed as the mean ± standard deviation.

Type	n	Weight (g)	Length (cm)	Width (cm)	Clasper Length (cm)
Aquarium-housed	7	309.9 ± 31.35	45.1 ± 1.88	11.3 ± 0.65 ^a^	3.9 ± 0.17
Wild-captured	17	269.5 ± 50.28	44.9 ± 3.09	10.9 ± 0.90 ^b^	3.7 ± 0.55
*p*-value		0.063	0.862	<0.001	0.265

Different superscript letters (a, b) in the same column indicate differences (*p* < 0.05).

**Table 2 animals-11-02884-t002:** Sperm quality comparison for aquarium-housed and wild-captured small-spotted catshark (*Scyliorhinus canicula*). Data are expressed as the mean ± standard deviation.

Semen Quality Variable	Aquarium-Housed	Wild-Captured	*p*-Value
Semen sample volume (mL)	0.8 ± 0.42 ^b^	1.9 ± 1.11 ^a^	0.005
Sperm concentration (10^6^/^mL^)	78.9 ± 43.36	107.8 ± 50.42	0.072
Total sperm (10^6^/sample)	72.7 ± 68.27 ^b^	219.6 ± 188.57 ^a^	0.006
Motility (%)	48.7 ± 6.96 ^a^	18.8 ± 4.1 ^b^	0.004
Viability (%)	72.9 ± 5.55	78.3 ± 3.30	0.717
Mitochondrial membrane high potential (%)	90.1 ± 6.97	96.1 ± 0.70	0.372
Sperm membrane integrity (%)	70.0 ± 6.75	72.9 ± 5.14	0.882
Number of animals (total samples)	7 (18)	17 (17)	

Different superscript letters (a, b) in the same row indicate differences (*p* < 0.05).

**Table 3 animals-11-02884-t003:** Spermatozoa morphometric comparison for aquarium-housed and wild-captured small-spotted catshark (*Scyliorhinus canicula*). Data are expressed as the mean ± standard deviation and coefficient of variation (CV, %).

Type	Acrosome (μm)	Head (μm)	Acrosome & Head (μm)	Midpiece (μm)
Aquarium-housed (CV)	2.78 ± 0.466 ^a^ (16.74)	40.96 ± 1.113 ^a^ (2.71)	43.75 ± 1.153 ^a^ (2.63)	18.16 ± 0.988 (5.44)
Wild-captured (CV)	3.01 ± 0.354 ^b^ (11.80)	41.79 ± 1.284 ^b^ (3.07)	44.80 ± 1.274 ^b^ (2.84)	18.31 ± 1.139 (6.22)
*p*-value	0.027	0.005	0.001	0.566

Different superscript letters (a, b) in the same column indicate differences (*p* < 0.05).

## Data Availability

The data used to support the findings of this study are included within the article.

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
