# Peer review of "Comparative Study of Semen Parameters and Hormone Profile in Small-Spotted Catshark (Scyliorhinus canicula): Aquarium-Housed vs. Wild-Captured"

_animals, 2021, doi:10.3390/ani11102884_

Round 1
Reviewer 1 Report
Comparative study of semen parameters and hormone profile in small-spotted catshark (Scyliorhinus canicula): aquarium-housed vs wild-captured is a nice paper focused on evaluation of suitable males for reproduction from captivity in comparison with males recently caught from the sea. I recommend publishing the article after the minor revision.
Line 88-89 Clearly write when the sharks were caught from the sea, describe their history of where they came from and what their growth was in captivity. Make it clear that these are wild individuals who have been successfully adapted for captivity.
Table 2 Adjust the seed volume to ml instead of µl.
Author Response
Reviewer 1 wrote: Comparative study of semen parameters and hormone profile in small-spotted catshark (Scyliorhinus canicula): aquarium-housed vs wild-captured is a nice paper focused on evaluation of suitable males for reproduction from captivity in comparison with males recently caught from the sea. I recommend publishing the article after the minor revision.
Line 88-89 Clearly write when the sharks were caught from the sea, describe their history of where they came from and what their growth was in captivity. Make it clear that these are wild individuals who have been successfully adapted for captivity.
Response to Reviewer 1
The authors would like to thank the referee for his/her careful reading of the manuscript and the comments. We want to thank the reviewer for pointing out minor spell checks. We have carefully edited this new version and corrected them.
The description of aquarium shark history has been completed in the highlighted manuscript in section 2.3, Sample collection and processing.
“The aquarium-housed animals were maintained under human care at the aquarium at least one year before the experiment started, being completely adapted to the aquarium environment”.
Moreover, the biometric study showed no significative differences between both groups. That is the reason why the authors assume the successful adaptation to captivity.
The description of wild shark history has been also completed in the highlighted manuscript in the same section.
“In the case of wild-captured sharks, samples were taken immediately when the fishery boat arrived at the port, between 4-8 h post-capture in the sea, depending on the daily fisheries routines”.
Reviewer 1 wrote: Table 2 Adjust the seed volume to ml instead of µl.
Response to Reviewer 1
Thanks for the comment, the volume unit has been modified in Table 2.

Reviewer 2 Report
Please see comments on the methodology especially about collecting samples from dead sharks and this means problems when comparing it to live-shark samples, due to post mortem degeneration and changes.

Author Response
Reviewer 2 wrote: Please see comments on the methodology especially about collecting samples from dead sharks and this means problems when comparing it to live-shark samples, due to postmortem degeneration and changes.
Response to Reviewer 2
The authors would like to thank the referee for his/her careful reading of the manuscript and the comments. We want to thank the reviewer for pointing out minor spell checks and gap spaces. We have carefully edited this new version and corrected them according to the reviewer´s suggestions. Specific responses of each comment are above.
Reviewer 2 wrote: Line 40-41. Discuss why this may be so. Endocrine disruption in sharks? See https://www.researchgate.net/publication/7060109_Endocrine_disruptors_in_marine_organisms_Approaches_and_perspectives
Response to Reviewer 2
Thanks for the comment, the description of endocrine disruption effects in effects in shark reproduction has been expanded in the highlighted manuscript.
“Of the 1226 species of sharks and rays [7] and 52 chimaeras species [8] in the wild, it is estimated that more than one quarter are threatened according to IUCN Red List criteria, due to overfishing (targeted and incidental), habitat degradation and other pollution threads such as heavy metals, microplastics or endocrine disruptors [9–13]”.
Reviewer 2 wrote: Line 53. Expand the discussion on pollution effects. See: https://www.sciencedirect.com/science/article/pii/S0025326X20308195
https://www.vin.com/apputil/content/defaultadv1.aspxpId=11257&catId=32286&id=3864677
Response to Reviewer 2
Thanks for the comment, the description of pollution effects in sharks physiology has been expanded in the highlighted manuscript (described in the previous response). However, the main objective of this study is not compare the pollution effects on these animals, authors will take this consideration for further studies.
Reviewer 2 wrote: Line 114-116. Results need to be placed in the results section.
Response to Reviewer 2
Thanks for the comment, the biometric information has been transferred to the result section.
Reviewer 2 wrote: Line 146. How long after death in the shark was the semen or blood sample collected?
See: https://www.ncbi.nlm.nih.gov/pmc/articles/PMC5196027/
About the quality of semen collected from a dead animal, it has to be quick - very soon after slaughter to maintain quality.
Example = "The testicles were removed 15–30 min after slaughter and transported at 18–20°C for ~1 h to the laboratory." from above paper.
Response to Reviewer 2
Authors decided to collect the samples from recent dead wild-type sharks due to the impossibility of getting live S. canicula in the wild. Local fisheries capture these sharks by accident (by-catch) using trawling methods during their daily fisheries routines, and the authors only have access to them once they reach the port. The authors were not allowed to be on board in the fishing boat during their activities. For that reason, semen and blood were collected immediately after the arrival to the port, between 4-8 h after capture, depending on the daily fisheries routine. In all cases, animals had signs of recent dead. Samples were maintained in the dark at 4ºC and then transferred within one hour to the laboratory facilities, for the immediate assessment. Same conditions were applied for aquarium animal’s samples.
We are grateful for the reviewer's suggestion, and we have added some extra information in the manuscript to clarify this point.
“In the case of wild-captured sharks, samples were taken immediately when the fishery boat arrived at the port, between 4-8 h post-capture in the sea, depending on the daily fisheries routines”.
Reviewer 2 wrote: Line 148. Blood from dead sharks not suitable for analysis due to postmortem artifacts, so generally shark reproductive studies collect blood samples from living sharks
See: https://academic.oup.com/conphys/article/2/1/cou013/327169
https://www.sciencedirect.com/science/article/pii/S0378432018304226?via%3Dihub
Response to Reviewer 2
As mentioned before, the blood collection from wild animals recently captured was not possible in our conditions, due to the difficulty obtaining live sharks from the wild. However, the samples were obtained in a short period after caught (4-8h) and immediately processed. In fact, the testosterone levels obtained in wild animals were statistically higher than their captive counterpart. This fact agrees with previous reports in another shark species, where testosterone levels were compared between captive and wild sharks, both in living animals (Wyffels et at., 2020). Moreover, other authors have used dead organism to obtain and analyse different hormones, mainly in commercial species (Becerril-García et al. 2020), including those analysed in the present paper.
Reviewer 2 wrote: Line 228. Discuss how the hormones can degrade after an animal has died.
See: https://pubmed.ncbi.nlm.nih.gov/8706491/
That is why for comparison, the blood samples need to be taken from a live animal so as to be accurate and a reflection of the status in the living animal.
Your wild shark-dead samples likely cannot be compared to your aquarium live-shark samples.
Response to Reviewer 2
We thank the reviewer for the comment. However, we could not access to wild live animals for blood collection. The authors obtained the blood samples immediately after the animals arrival to port after catch. Several authors contemplate lethal techniques, considering it useful for sexual hormonal analysis (retrospective study performed by Becerril-García et at., 2020).
In our results, testosterone levels was higher in wild animals compared with aquarium sharks, similarly to other authors (Wyffels et al,.2020) comparing living C. taurus in the wild and in aquaria. The effects of stress induced during by-catch due to the elevation of plasma cortisol and posterior dead, could affect, or even suppress circulating androgens in other teleost fishes (Pickering et al., 1987), but according to our results, and even considering the possible sexual hormonal degradations, the differences between groups still exist in favour of wild animals.
The hormonal discussion has been completed in the highlighted manuscript;
“The effect of cortisol produced during the capture and death process could decrease androgen levels in wild fish [60], although other research used samples from commercial sharks species after lethal techniques to analyze sexual hormones [61]. The present study found higher testosterone levels in wild-captured sharks similarly to other results reported in previous research using living animals [21]”.
Reviewer 2 wrote: Line 277, Table 2, about semen sample volume. When the shark has died, the blood stops flowing and tissue fluid builds up which can affect the quantity and quality of the sperm. Look at the very low motility % - which is a key indicator of sperm quality.
Response to Reviewer 2
Thanks for the comments. The quality expressed in motility could be related to the death, but in this study, the rest of parameters (viability, mitochondrial membrane potential or membrane integrity) do not show differences between groups. For that reason, we consider that the early post-mortem sample collection does not affect the sperm quality. The motility discussion has been completed in the highlighted manuscript;
“Despite those semen peculiarities related with wild-captured sharks, sperm viability, mitochondrial membrane potential, and sperm membrane integrity were similar in aquarium-housed and wild-captured small-spotted catsharks, even after considering the possible effect of sample collection in recent deceased animals”.
In terms of semen quantity samples, wild sharks showed higher sperm volumes. This fact could be associated with the muscle relaxation of deceased animals, making possible a complete collection of semen from the ampulla. Another possible explanation could be related to higher levels of sexual competition in wild sharks.
This information is described in the manuscript:
“In wild-captured animals, the semen was collected from recently dead animals, the by-catch by trawling (from 4 to 8 hours after death depending on the fisheries daily routines), where muscles were relaxed, and the ampulla could be emptied. Abdominal massage or stripping, as occurs in aquarium-housed sharks, only releases mature spermatozoa together with seminal plasma [46], but when this technique is used in freshly dead sharks, it is possible to recover the vast majority of sperm present in the caudal reproductive tract. This fact could also explain the differences in motility between groups as wild-type semen samples comprised all the sperm forms contained on the ducts. Another alternative and potentially complementary explanation for the higher semen volume observed may be the sexual competition between wild-captured males and the post-copulatory sperm competition due to female mating attempts with multiple males [21,47]”.
Reviewer 2 wrote: Line 277, Table 2, about total sperm count. This may be due to the sharks being dead for long enough so that the sperm is starting to degrade.
Response to Reviewer 2
We thank the reviewer for the comment. However, in this case we do not agree with it. The higher volume and the total number of sperm observed in the results are related to wild animals, not to housed-sharks, so the sperm are not being degraded because of the animal deceased.
Reviewer 2 wrote: Line 360. Likely to have postmortem changes that can alter the data on sperm quality and hormone levels.
Response to Reviewer 2
The responses to this question have been already described previously.
Reviewer 2 wrote: Line 422. Need to repeat the study using live wild shark samples.
Response to Reviewer 2
We take into account the referee´s comment for future research. The authors are aware that the ideal experimental design is to compare live animals in both conditions (wild vs. captive); however, the procedure to obtain wild animals for this study were hardly complicated because we could not have any other option that wait for fisheries at the port due to the impossibility to be on board with them and obtain the samples immediately after catch. The authors consider that the experimental procedures are well done considering these limitations that are already mentioned in the discussion section. The authors consider that the present study is of special interest due to it is the first comparison between reproductive parameters in this species, opening new insights for further studies in shark conservation.

Reviewer 3 Report
The study is the valuable data to present semen quality and the hormone profile of small-spotted catsharks between wild-captured and aquarium-housed.
The purpose in this study is clearly described and experimental design including methods is well-organized. However, one thing lacking is the assay of motility. You could get more precise results by using CASA instead of subjective observation, though you cited the reference about measurement of sperm motility. The result of motility is one key point of this work. The other thing is you should mention the ethical approval by the committee.
Author Response
Reviewer 3 wrote: The study is the valuable data to present semen quality and the hormone profile of small-spotted catsharks between wild-captured and aquarium-housed.
The purpose in this study is clearly described and experimental design including methods is well-organized. However, one thing lacking is the assay of motility. You could get more precise results by using CASA instead of subjective observation, though you cited the reference about measurement of sperm motility. The result of motility is one key point of this work.
Response to Reviewer 3
The authors would like to thank the referee for his/her careful reading of the manuscript and the kind comments.
Actually, CASA technology has been used in other chondrycthyans sperm research (Dzyuba et al., 2019; Penfold et al., 2019), however we did not have yet set up this system for S. canicula spermatozoa. We are working on it for further research in this species.
Reviewer 3 wrote: The other thing is you should mention the ethical approval by the committee.
Response to Reviewer 3
This information is included in the Institutional Review Board Statement of the manuscript. “All the procedures involving aquarium-housed animals in this study were approved by the Animal Care and Welfare Committee of the OceanograÌ€fic Valencia (Reference number: OCE-18-19) following the Animal Care Protocol and policies of the aquarium. In the case of wild animals, they were fresh accidental captures with commercial value, donated by local fisheries”.

Reviewer 4 Report
The paper is well written and the results are clearly presented.
The only concern that I have is "why authors decided to collect sperm from dead wild-type catfish as it may have affected the sperm quality and hormone levels significantly?" I believe wild-type subjects also should be treated similarly for samples collections, hence keeping the treatment the same for all parameters.
Thanks,
Author Response
Reviewer 4 wrote: The paper is well written and the results are clearly presented.
The only concern that I have is "why authors decided to collect sperm from dead wild-type catfish as it may have affected the sperm quality and hormone levels significantly?" I believe wild-type subjects also should be treated similarly for samples collections, hence keeping the treatment the same for all parameters.
Response to Reviewer 4
The authors would like to thank the referee for his/her careful reading of the manuscript and the comments. We want to thank the reviewer for pointing out minor spell checks. We have carefully edited this new version and corrected them.
Authors decided to collect the samples from recent dead wild-type sharks due to the impossibility of getting live S. canicula in the wild. Local fisheries capture these sharks by accident (by-catch) using trawling methods during their daily fisheries routines, and the authors only have access to them once they reach the port. The authors were not allowed to be on board in the fishing boat during their activities. For that reason, semen and blood were collected immediately after the arrival to the port, between 4-8 h after capture, depending on the daily fisheries routine. In all cases, animals had signs of recent dead. Samples were maintained in the dark at 4ºC and then transferred to the laboratory facilities for the immediate assessment. Same conditions were applied for aquarium animal’s samples.
In fact, most of the sperm quality parameters analysed in the present study (viability, mitochondrial membrane potential or membrane integrity) were similar between both groups of animals. We agree with the reviewer that the low level of motile sperm observed in wild animals could be related to to death. This information has been included in the discussion section. For these reasons, we consider that the early post-mortem sample collection, although no-ideal, is a very fine approach to compare with live animals Moreover, the testosterone levels obtained in wild animals were statistically higher than their captive counterpart. This fact agrees with previous reports in another shark species, where testosterone levels were compared between captive and wild sharks, both in living animals (Wyffels et at., 2020). Furthermore, other authors have used dead organism to obtain and analyse different hormones, mainly in commercial species (Becerril-García et al. 2020), including those analysed in the present paper. This description has been expanded in the highlighted manuscript;
“The effect of cortisol produced during the capture and death process could decrease androgen levels in wild fish [60], although other research used samples from commercial sharks species after lethal techniques to analyze sexual hormones [61]. The present study found higher testosterone levels in wild-captured sharks similarly to other results reported in previous research using living animals [21]”.
The authors consider that the present study is of special interest due to it is the first comparison between reproductive parameters in this species, opening new insights for further studies in shark conservation.

Round 2
Reviewer 2 Report
Thank you for addressing the questions, unfortunately the samples are best taken from both captive and wild live sharks. For wild sharks it is difficult using the fish trawler system, but for the validity of the results in all aspects, special effort to sample from live wild sharks is best.
Author Response
Thank you for your comments. We are aware of the limitations of the experimental design but in our Country, the procedure to obtain live wild animals was not possible currently. Nevertheless, we consider important for chondrichthyan conservation the possibility of gametes evaluation from newly dead animals when there are no possibilities of getting that information from live animals.